# Study on the Concentration of Acrylic Acid and Acetic Acid by Reverse Osmosis

**DOI:** 10.3390/membranes10070142

**Published:** 2020-07-06

**Authors:** Qian Liu, Lixin Xie, Hanxiao Du, Shichang Xu, Yawei Du

**Affiliations:** 1School of Chemical Engineering and Technology, Tianjin University, Tianjin 300072, China; lq202071@126.com (Q.L.); d826258049@126.com (H.D.); xushichang@sina.com (S.X.); 2Tianjin Key Laboratory of Membrane Science and Desalination Technology, State Key Laboratory of Chemical Engineering (Tianjin University), Tianjin 300072, China; 3School of Chemical Engineering and Technology, Hebei University of Technology, Tianjin 300130, China

**Keywords:** reverse osmosis (RO), concentration, acrylic acid, acetic acid, membrane stability

## Abstract

In the production of acrylic acid, the concentration of acrylic acid solution from the adsorption tower was low, which would lead to significant energy consumption in the distillation process to purify acrylic acid, along with the production of a large amount of wastewater. Reverse osmosis (RO) was proposed to concentrate the acrylic acid aqueous solution taken from a specific tray in the absorption tower. The effects of operating conditions on the permeate flux and acid retention were studied with two commercial RO membranes (SWC5 and SWC6). When the operating pressure was 4 MPa and the temperature was 25 °C, the permeate fluxes of two membranes were about 20 L·m^−2^·h^−1^. The acrylic acid and acetic acid retentions were about 80% and 78%, respectively. After being immersed in the acid solutions for several months, the characteristics of the two membranes were tested to evaluate their acid resistance. After six months of exposure to the acid solution containing 2.5% acrylic acid and 2.5% acetic acid, the retentions of acrylic acid and acetic acid were decreased by 5.7% and 4.1% for SWC5 and 4.9% and 2.2% for SWC6, respectively. The changes of membrane surface morphology and chemical composition showed the hydrolysis of some amide bonds.

## 1. Introduction

Acrylic acid (AA) is a versatile monomer which is widely used in the synthesis of plastics, synthetic rubbers, superabsorbent polymers, coatings, detergents, fibers, and specialty resins [1,2,3,4,5]. Currently, the main production technology of acrylic acid is the two-step gas-phase oxidation of propylene [1,2,3]. The mixed gases produced in the oxidation process are introduced into the bottom of the absorption tower. Acrylic acid and multiple by-products, including acetic acid (HAc), formic acid, maleic acid, acrolein, acetaldehyde acetone, etc. [6], are absorbed by the water sprayed down from the top of absorption tower. Then, the acrylic acid aqueous solution can be obtained at the bottom. Limited by the high concentration of water vapor at the inlet of the oxidation reactor and the absorption method, the concentration of acrylic acid in the bottom of the absorption tower is relatively low, which results in a large amount of energy consumption and wastewater production in downstream distillation process [7]. Some methods were proposed to improve the concentration of the acrylic acid solution before the distillation process. Briegel et al. [8] used a condenser tower equipped with multiple external heat exchangers instead of the conventional absorption tower to indirectly cool and recover a higher concentration of (meth)acrylic acid solution from the gaseous steam. Min et al. [9] used the extraction solvent to extract (meth)acrylic acid from the side stream of the absorption tower. Although the concentration of (meth)acrylic acid solution entering the distillation tower was increased, the large amount of extraction solvent was inevitable.

In recent years, membrane technologies, including microfiltration (MF), ultrafiltration (UF), nanofiltration (NF), and reverse osmosis (RO), have attracted great attention during the industrial production process [10,11,12,13]. With the development of a variety of high-performance RO membranes, such as antifouling membranes, acid resistance membranes, ultra-low pressure membranes, and high retention membranes, the application of RO has gradually expanded from seawater desalination to the fields of wastewater treatment [14], food processing [15], petrochemical industry [16], pharmaceutical industry [17], and acid concentration and separation [14,18]. Ricci et al. [19] integrated NF and RO to separate noble metals ions and concentrate sulfuric acid from the gold mining effluent. The rejection of metals ions by the NF membrane was above 90%, and the sulfuric acid could permeate through the membrane. With the recovery of 50%, an increase of 99% in sulfuric acid concentration compared to the feed was achieved. González et al. [18] determined the feasibility of purifying industrial phosphoric acid solution by RO. Moreover, the retention of cationic impurities were 99.3%, and 46.3% of acid permeation were achieved. Zhou et al. [20] separated acetic acid from model lignocellulosic hydrolysates by RO. It was shown that the separation factor of acetic acid over sugars was above 200. Chen et al. [21] developed an ethanol promotion method to facilitate the removal of furfural and acetic acid from hydrolysate. When the ratio of ethanol/acetic acid concentration was 5.80, the acetic acid retention of PA2-4040 RO membrane was only 10%. Ahsan [22] reported that multi-stage RO could recover about 70% acetic acid from prehydrolysis liquor of kraft. Tan et al. [23] immersed the SWC5 and SWC6 membranes in the hydrochloric acid solution with a pH value of 1 for 2 h. Although the hydrolysis of some amide bonds of the treated membranes was observed, the salt retention of the two membranes were decreased by less than 1%. These studies indicated that the RO had great potential in the concentration and separation of acid. However, the above studies did not assess the long-term stability of the membrane in the specific acid solution.

In this study, RO was proposed to concentrate the acrylic acid aqueous solution taken from a specific tray in the absorption tower. The concentrated acrylic acid solution was returned to the absorption tower for continuous absorption. The permeate from RO was reused as absorbent. The utilization of the process would decrease the amount of fresh absorbent and increase the concentration of acrylic acid in the bottom of the absorption tower, thereby decreasing the amount of wastewater production and energy consumption in the distillation process.

The feasibility of using RO to concentrate the synthetic solution containing acrylic acid and acetic acid was evaluated in this paper. The effects of operating pressure, temperature and feed concentration on permeate flux and acid retention were investigated. The stability of RO membrane was also tested by detecting the membrane characteristics after continuous exposure to the acid solution.

## 2. Materials and Methods

### 2.1. Materials

Two commercial seawater RO membranes (SWC5 and SWC6) were acquired from Hydranautics, Oceanside, CA, USA, which were marked as C5 and C6 in this study, respectively. Table 1 showed the specifications of the two membranes.

Acetic acid (≥99.5%) was purchased from Tianjin Kemel Co., Ltd., Tianjin, China. Acrylic acid (≥99.5%) and sodium chloride (≥99.5%) were purchased from Tianjin Damao Co., Ltd., Tianjin, China. Anhydrous grade ethanol was obtained from Aladdin Reagent Co., Ltd., Shanghai, China. 

### 2.2. Filtration Experiment

All filtration experiments were carried out by a lab-made cross-flow RO filtration apparatus, as illustrated in Figure 1. The apparatus mainly consisted of feed tank, heat exchanger, high-pressure pump, membrane cells, pressure gauges, and rotameter. The effective area of the membrane cell was 31.16 cm^2^. 

The permeate flux and acid retention of the RO membranes were tested under different operating conditions (pressure, temperature, and feed concentration). The operating pressure was adjusted by the valve on the outlet of the concentrate. The heat exchanger was used to control the solution temperature. The system was operated under a recirculating flow rate of 1.5 L·min^−1^. The permeate flux was obtained by measuring the volume of permeate over a period of time, and the acid retention was determined through analyzing the acid concentrations of permeate and feed. These measurements were repeated three times under the same experimental conditions. The permeate flux and acid retention were calculated by Equations (1) and (2), respectively.
(1)$$J=\frac{V}{A\Delta t}$$
(2)$$R_{\mathrm{acid}}\left(\%\right)=\left(1-\frac{C_{p}}{C_{f}}\right)\times 100$$
where J is the permeate flux (L·m^−2^·h^−1^), V is the permeate volume (L), A is the effective membrane area (m^2^), and ∆t is the measuring time (h). R_acid_ is the acid retention and C_p_ and C_f_ are the acid concentrations of the permeate and feed (g·L^−1^), respectively.

### 2.3. Immersion Experiment

To investigate the effects of continuous exposure to the acid solutions on the membrane characteristics, the two RO membranes (C5 and C6) were immersed in the acid solutions (Table 2) for 6 months at room temperature. At two months intervals, the membrane samples were taken and washed with deionized (DI) water. Subsequently, they were used for acid filtration experiments (Section 2.2), and their morphological and chemical characteristics were also tested (Section 2.4).

### 2.4. Analytical Methods

The pH value of the solution was tested by a pH-meter (FE28-Standard, METTLER TOLEDO, Switzerland). The concentrations of acrylic acid and acetic acid were measured by Gas chromatography (GC, SP-2100A, Beifen, Beijing, China) with a FID detector and a capillary column (KB-FFAP, 30 m × 0.32 mm × 0.5 μm) using ethanol as an internal standard [24,25]. The temperatures of the injector, detector, and column were 200 °C, 220 °C, and 150 °C, respectively. The flow rates of N_2_, air and H_2_ were 20 mL/min, 300 mL/min and 30 mL/min, respectively.

The membrane surface morphology was analyzed by scanning electron microscopy (SEM, S4800, Hitachi, Tokyo, Japan). The roughness of the membrane surface was tested by atomic force microscopy (AFM, Dimension icon, Bruker, Karlsruhe, Germany) using tapping mode. The membrane surface chemical composition was characterized by Fourier transform infrared spectroscopy (FT–IR, 6700, Nicolet, Madison, WI, USA) and X-ray photoelectron spectroscopy (XPS, ESCALAB-250Xi, ThermoFisher, Waltham, MA, USA).

## 3. Results and Discussion

### 3.1. The Concentration Performance of the RO Membranes

#### 3.1.1. Effect of Pressure

The effect of pressure on permeate flux and acid retention for C5 and C6 at 25 °C was showed in Figure 2. The feed solution contained 2.5% acrylic acid and 1.5% acetic acid based on the concentration of a specific tray in the absorption tower of the acrylic acid production process. The pH value of the feed solution was 2.39. 

As shown in Figure 2a, the permeate fluxes of both membranes were linearly dependent on the pressure. When the pressure varied from 2.0 MPa to 4.0 MPa, the permeate fluxes were increased from 7.89 L·m^−2^·h^−1^ to 19.69 L·m^−2^·h^−1^ for C5, and 8.36 L·m^−2^·h^−1^ to 21.12 L·m^−2^·h^−1^ for C6, respectively. Meanwhile, the retentions of acrylic acid (Figure 2b) were significantly increased from 67.40% to 81.92% for C5, and 66.73% to 81.32% for C6, respectively. The retentions of acetic acid were slightly lower than those of acrylic acid. The similar trend was also reported by Zhou et al. [20,26] during the separation acetic acid from monosaccharides by RO. This phenomenon was induced by solution-diffusion theory [27]. With the increase of pressure, the water flux increased faster than solute flux, so the solute retention was increased. In addition, the solution-diffusion model was used to calculate the water permeability and the transport coefficients of two acids [28], as shown in Appendix A. For the acid solution containing 2.5% acrylic acid and 1.5% acetic acid, the water permeability coefficients of C5 and C6 were 7.195 L·m^−2^·h^−1^·MPa^−1^ and 7.694 L·m^−2^·h^−1^·MPa^−1^, respectively. The transport coefficients of acetic acid and acrylic acid were 4.275 L·m^−2^·h^−1^ and 3.921 L·m^−2^·h^−1^ for C5, and 4.669 L·m^−2^·h^−1^ and 4.272 L·m^−2^·h^−1^ for C6, respectively. 

As shown in Figure 2b, the retentions of acrylic acid and acetic acid were significantly lower than that of NaCl (Table 1). The similar results were also reported in the previous study [20,27,29,30], as shown in Appendix A. The lower acid retention may be attributed to the following two factors. According to the surface absorption theory, the surface tension of the solution containing acrylic acid and acetic acid was lower than that of NaCl solution [31,32], which would cause more acid absorption by the membrane. On the other hand, the hydrogen bonding between the acid (AA and HAc) and the PA membrane would also increase the acid absorption [33]. NaCl was a strong electrolyte and existed as Na^+^ and Cl^−^ in the solution. The electrostatic repulsion between the PA membrane and the charged ions would increase the retention of NaCl [34]. Thus, NaCl could be better retained than acetic acid and acrylic acid by the membrane. The difference in hydrophilicity may be one of the reasons for the different retention of the two acids. The hydrophilicity of a compound was often described by the octanol/water partitioning coefficient (log(K_ow_)), where a lower log(K_ow_) was corresponding to a more hydrophilic compound [35,36]. So, the acetic acid (log(K_ow_) = −0.17) [37] with a lower log(K_ow_) was more permeable than acrylic acid (log(K_ow_) = 0.36) at the same condition [35]. 

#### 3.1.2. Effect of Temperature

The effect of temperature on permeate flux and acid retention was investigated from 20 °C to 35 °C at 3.5 MPa for C5 and C6. The feed concentrations of acrylic acid and acetic acid were 2.5% and 1.5%, respectively. The permeate flux (Figure 3a) was increased almost linearly as temperature increased. However, the retentions of acrylic acid and acetic acid (Figure 3b) were significantly declined. As the temperature increased from 20 °C to 35 °C, the retentions of acrylic acid and acetic acid (Figure 3a) were decreased by 9.6% and 9.5% for C5, and 12.4% and 11.5% for C6, respectively.

According to the Arrhenius relation, increasing temperature would promote the transport of water and solute through the membrane due to the increase of diffusion coefficient [38]. Moreover, it was reported that the increase in the mass transfer of solute was more significant than that of water with the increase of temperature [39], which would increase the permeate flux and reduce acid retention. The decrease of acid retention may be also attributed to the increase in membrane pore size caused by the thermal dilation of the polymer in the active layer at higher temperature [38,40].

In the absorption process of acrylic acid production, the temperatures in the top and bottom of the absorption tower are about 43 °C and 70 °C, respectively. The temperature of the solution taken from the absorption tower was about 45 °C. According to the operation conditions of the membrane recommended by the manufacturer, the maximum operating temperature of the two RO membranes is 45 °C. Therefore, the solution taken from the absorption tower was required to cool down before entering into the RO unit.

#### 3.1.3. Effect of Feed Concentration

The effect of feed concentration on the RO membrane performance was assessed at 3.5 MPa, 25 °C with two groups of acid solutions. In group A, the concentration of acrylic acid was 1.5%, 2%, 2.5%, and 3%, respectively, with the 1.5% acetic acid. In group B, the concentration of acetic acid was 1%, 1.5%, 2%, and 2.5%, respectively, with the 2.5% acrylic acid. 

Figure 4a,c showed that the permeate fluxes of the two membranes were gradually decreased with the increase of the concentration of acrylic acid or acetic acid. The decrease in permeate flux was mainly due to the reduction in effective pressure caused by the increase of osmosis pressure [14,41]. There was no discernable difference in the retentions of acrylic acid and acetic acid in both the two membranes (Figure 4b,d). The similar results were also reported by other researchers [20,29]. For both the two membranes, the retentions of acrylic acid and acetic acid were about 80% and 78%, respectively.

### 3.2. Effect of Continuous Exposure to the Acid Solutions on the Performance and Characteristics of RO Membranes

#### 3.2.1. Permeate Flux and Acid Retention

After immersed in the acidic solutions for several months, the permeate flux and acid retention of the RO membranes were tested to evaluate their acid resistance. The filtration experiments were performed at 3.5 MPa and 25 °C with an acid solution containing 2.5% acrylic acid and 1.5% acetic acid. 

As shown in Figure 5, the permeate flux was increased, and the acid retention was decreased with the increase of the exposure time for all samples. The influences were more evident with the higher concentration of acid solution. The permeate fluxes of the samples C5-2.5 and C6-2.5 were increased by 33.1% and 15.3% after six months, respectively. The acrylic acid retentions of those were decreased by 5.7% and 4.9%, respectively. Moreover, the acetic acid retentions of the two samples were decreased by 4.1% and 2.2%, respectively. At the end of six months of exposure, the performance of samples C5-7.5 and C6-7.5 was degraded significantly. Compared with the virgin membranes, the retentions of acrylic acid and acetic acid were decreased by 14.4% and 9.6% for sample C5-7.5, and 11.7% and 6.4% for sample C6-7.5, respectively.

#### 3.2.2. Membrane Surface Morphology

The surface morphology of C5-2.5, C5-7.5, C6-2.5 and C6-7.5 before and after being immersed for six months was investigated by SEM. As shown in Figure 6, all samples exhibited a typical ridge-and-valley structure of the PA membrane. There was no apparent difference between the ridge structures of treated samples C5-2.5 and C6-2.5 and the untreated ones (samples C5-Virgin and C6-Virgin). The ridge structures of samples C5-7.5 and C6-7.5 spread out gradually attributed to the membranes swelling in the acid solutions [42]. In addition, as the acid concentration increased, the amount of ridge structures became less, and the area of each ridge structure became larger. 

The roughness of samples C5-2.5, C5-7.5, C6-2.5 and C6-7.5 after exposure for 0, 4, 6 months was tested by AFM, as shown in Table 3. The AFM images of the RO membranes were presented in Appendix A. It was observed that the average roughness of all samples was decreased gradually with the exposure time increasing. For example, at the end of six months, the roughness of samples C5-7.5 and C6-7.5 was decreased by 24.8% and 12.6%, respectively. The results indicated that the membrane surface became smoother, which was in accordance with the SEM micrographs. 

#### 3.2.3. Membrane Surface Chemical Composition

The effect of continuous exposure to the acid solutions on membrane surface chemical composition was characterized by FT–IR and XPS. The FT–IR spectra detected in the range of 800–3500 cm^−1^ were shown in Figure 7, which contained both the bands of the active layer (PA) and the support layer (polysulfone, PSU) [43]. Two additional peaks were observed at 3300 cm^−1^ (NH^+^ and OH^−^ groups) [44] and 1723 cm^−1^ (C=O of carboxylic acid) [45] for the samples C5-2.5 and C6-2.5 after exposure for six months, which may be attributed to the following two factors. Firstly, the oxygen was more electronegative than carbon. Therefore, the electro cloud of oxygen was higher than that of carbon in the acid solutions [46]. The carbon was more vulnerable to nucleophilic attack when the H^+^ attacked the amide bond, leading to the hydrolysis of the amide bond to the-NH_2_ and -COOH [47]. Secondly, the acrylic acid and acetic acid in the acid solutions were easy to be absorbed onto the RO membranes, which also could result in the appearance of the peak at 1723 cm^−1^. However, no significant disappearance of the peak was observed after six months of exposure to the acid solutions. 

The surface chemical composition of samples C5-2.5, C5-7.5, C6-2.5 and C6-7.5 prior and after 6 months to exposure was also detected by XPS. The atomic percent of C1s (284.8 eV), O1s (531.3 eV), N1s (399.8 eV) and the atomic ratio of C/O, O/N were shown in Table 4. The atomic percent of carbon and oxygen was increased which of nitrogen was decreased attributed to the absorption of acrylic acid and acetic acid and the hydrolysis of the amide bond [44]. The atomic ratio of C/O was reduced and that of O/N was increased. The atomic ratio of O/N was 2 for the fully linear PA, while O/N was 1 for the fully cross-linked PA [43]. The increase of O/N suggested that there were more additional oxygen atoms without being bonded to nitrogen atoms. Consistent with the FT-IR results, the lower atomic ratio of C/O and the higher atomic ratio of O/N indicated more free carboxylic groups due to the hydrolysis of the amide bond [44]. 

Furthermore, the narrow scans of carbon and nitrogen were detected, as shown in Table 5. The corresponding C1s spectra and N1s spectra of the membranes before and after exposure were presented in Appendix A. There were four peaks of C1s in the results of all samples, which were observed at 284.3 eV (C–C, C–H), 285.1 eV (C–O, C–N), 287.7 eV (C=O, O=C–N), and 290.4 eV (π-π bonds) [44,48]. For nitrogen, the peak at 399.8 eV represented C–N and O=C–N [49]. Compared to the virgin membranes (samples C5-Virgin and C6-Virgin), an additional peak of the membranes exposed to acid solutions (samples C5-2.5, C5-7.5, C6-2.5 and C6-7.5) appeared in N1s scanning at 401.4 eV (–NH_3_^+^, –NH_2_R^+^). The component at 401.4 eV was increased with the increase of the acid solution concentration. It was indicated that a higher concentration of acidic solution could result in more severe degradation of the membranes. The results also verified the hydrolysis of the amide bond caused by acrylic acid and acetic acid.

## 4. Conclusions

In this work, two RO membranes were used to concentrate the solution containing acrylic acid and acetic acid under different operational conditions. With the pressure of 4 MPa and the temperature of 25 °C, the permeate fluxes of SWC5 and SWC6 were 19.69 L·m^−2^·h^−1^ and 21.12 L·m^−2^·h^−1^, respectively. For both membranes, the retentions of acrylic acid and acetic acid were around 80% and 78%, respectively. The stability of the membranes in the acid solutions was also assessed. The longer exposure time and the higher acid concentration would decline the membrane performance. After six months of exposure to the acid solution containing 2.5% acrylic acid and 2.5% acetic acid, the acrylic acid retentions of SWC5 and SWC6 were decreased by 5.7% and 4.9%, and the acetic acid retentions of those were decreased by 4.1% and 2.2%, respectively. With the acid solution containing 7.5% acrylic acid and 7.5% acetic acid, the retentions of acrylic acid and acetic acid were decreased by 14.4% and 9.6% for SWC5, and 11.7% and 6.4% for SWC6, respectively. The increase of membrane surface roughness and the hydrolysis of some amide bonds were consistent with the changes of membrane performance. The results showed that it is possible to concentrate a lower concentration side stream of the absorption tower by RO.

## Figures and Tables

**Figure 1 membranes-10-00142-f001:**
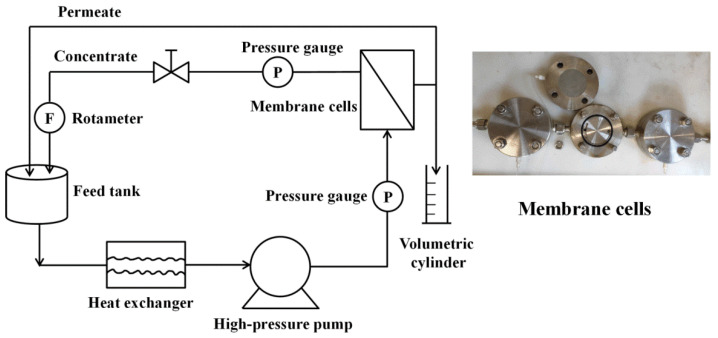
The schematic diagram of the cross-flow RO filtration apparatus.

**Figure 2 membranes-10-00142-f002:**
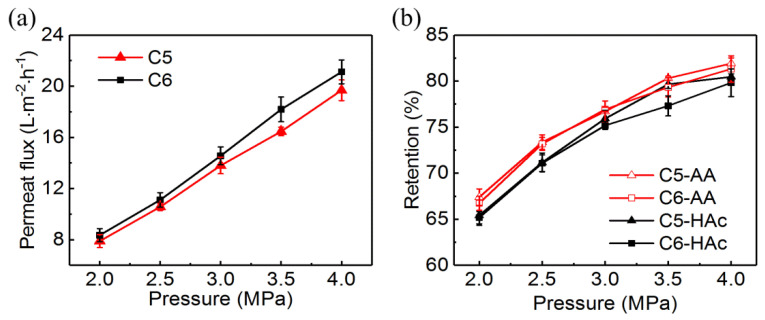
Effect of pressure on (**a**) permeate flux and (**b**) acrylic acid and acetic acid retentions. The concentrations of acrylic acid and acetic acid were 2.5%, 1.5%, respectively; temperature: 25 °C.

**Figure 3 membranes-10-00142-f003:**
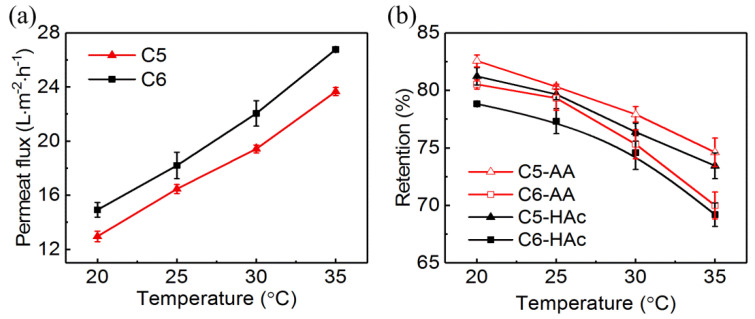
Effect of temperature on (**a**) permeate flux and (**b**) acrylic acid and acetic acid retentions. The concentrations of acrylic acid and acetic acid were 2.5%, 1.5%, respectively; pressure: 3.5 MPa.

**Figure 4 membranes-10-00142-f004:**
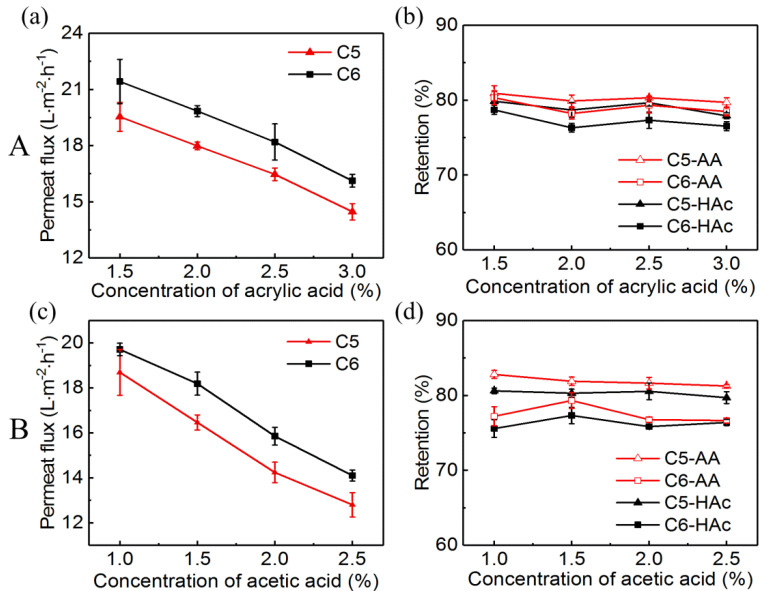
Effect of the feed concentration of acrylic acid (group **A**) on (**a**) permeate flux and (**b**) acrylic acid and acetic acid retentions, and the feed concentration of acetic acid (group **B**) on (**c**) permeate flux and (**d**) acrylic acid and acetic acid retentions. Pressure: 3.5 MPa; temperature: 25 °C.

**Figure 5 membranes-10-00142-f005:**
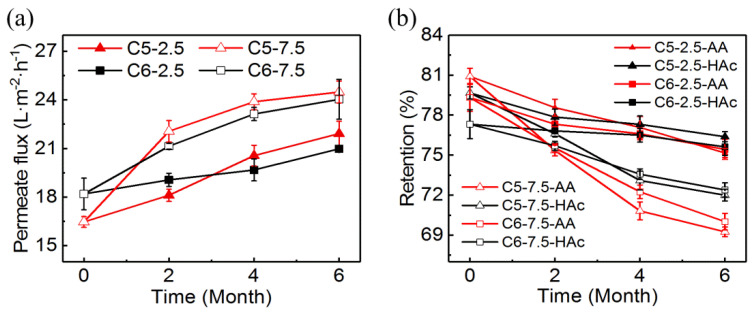
Effects of continuous exposure to the acid solutions on the (**a**) permeate flux and (**b**) acrylic acid and acetic acid retentions. The feed solution contained 2.5% acrylic acid and 1.5% acetic acid; temperature: 25 °C; pressure: 3.5 MPa.

**Figure 6 membranes-10-00142-f006:**
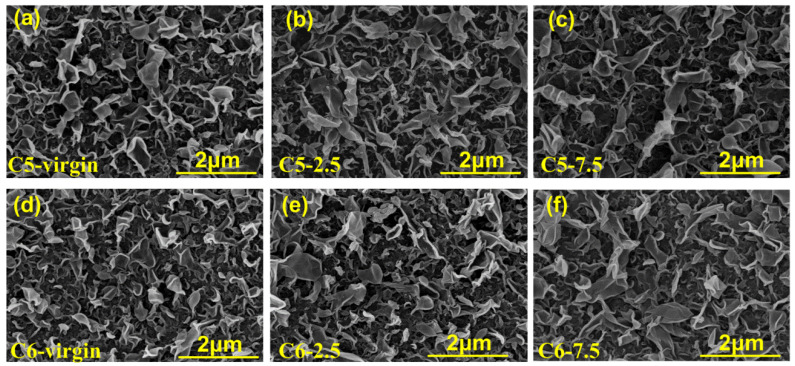
Surface SEM images of RO membranes: (**a**,**d**) prior to exposure and (**b**,**c**,**e**,**f**) after 6 months of exposure to the acid solutions.

**Figure 7 membranes-10-00142-f007:**
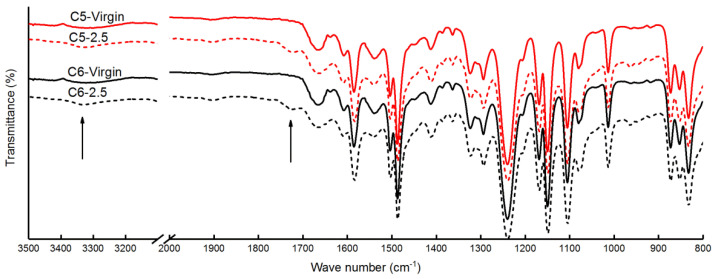
The FT–IR spectra of RO membranes before and after 6 months of exposure to the acid solutions containing 2.5% acrylic acid and 2.5% acetic acid.

**Table 1 membranes-10-00142-t001:** The specifications of the two RO membranes used in this study.

Membranes	C5	C6
Active layer	polyamide (PA)	PA
Operating pH	2–11	2–11
Maximum operating temperature (°C)	45	45
Maximum operating pressure (MPa)	8.27	8.27
NaCl retention (%)	99.8 ^a^	99.8 ^a^
Permeate flow rate (m^3^/d)	34.1 ^a^	45.4 ^a^

^a^ The test conditions specified by the manufacturer were: 32 g·L^−1^ NaCl solution at 5.52 MPa, 10% recovery, 25 °C and pH 6.5–7.0. The effective membrane area was 37.2 m^2^.

**Table 2 membranes-10-00142-t002:** Nomenclature and the acid solutions used for immersing the RO membranes.

Sample	Membrane	Solution Concentration
C5-Virgin ^a^	C5	-
C5-2.5	C5	2.5% AA + 2.5% HAc
C5-7.5	C5	7.5% AA + 7.5% HAc
C6-Virgin ^b^	C6	-
C6-2.5	C6	2.5% AA + 2.5% HAc
C6-7.5	C6	7.5% AA + 7.5% HAc

^a^ Sample C5-Virgin represented that samples C5-2.5 and C5-7.5 were immersed in the acid solutions for 0 months. ^b^ Sample C6-Virgin represented that samples C6-2.5 and C6-7.5 were immersed in the acid solutions for 0 months.

**Table 3 membranes-10-00142-t003:** The roughness of RO membranes before and after exposure to the acid solutions.

Sample	Average Roughness (nm)		
	0 Month	4 Months	6 Months
C5-2.5	137	127.5	117.0
C5-7.5	137	108.6	103
C6-2.5	87.2	86	78.9
C6-7.5	87.2	83	76.2

**Table 4 membranes-10-00142-t004:** The surface elemental composition of the RO membranes prior and after 6 months to exposure to the acid solutions by XPS.

Sample	Time (Months)	Atomic Percent (at. %)			Atomic Ratio	
		N	C	O	C/O	O/N
C5-Virgin	0	12.08	74.54	13.36	6.17	1.11
C5-2.5	6	10.27	75.43	14.30	5.27	1.39
C5-7.5	6	10.10	75.57	14.33	5.27	1.41
C6-Virgin	0	11.11	76.82	12.06	6.37	1.09
C6-2.5	6	10.76	76.26	12.99	5.87	1.21
C6-7.5	6	9.62	76.34	14.04	5.44	1.46

**Table 5 membranes-10-00142-t005:** The narrow scans of carbon (C1s) and nitrogen (N1s) of the RO membranes prior and after 6 months to exposure to the acid solutions by XPS.

Sample	Time (Months)	C 1s				N 1s	
		284.3 eV	285.1 eV	287.7 eV	290.4 eV	399.8 eV	401.4 eV
C5-Virgin	0	47.87	34.95	10.88	6.3	100	-
C5-2.5	6	49.14	33.44	11.23	6.18	99.04	0.96
C5-7.5	6	49.08	30.83	11.89	8.21	96.77	3.23
C6-Virgin	0	44.41	39.24	10.19	6.17	100	-
C6-2.5	6	48.19	32.7	11.43	7.67	98.94	1.06
C6-7.5	6	50.19	29.55	11.92	8.35	97.64	2.36

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
