# Peer review of "Study on the Concentration of Acrylic Acid and Acetic Acid by Reverse Osmosis"

_membranes, 2020, doi:10.3390/membranes10070142_

Round 1

Reviewer 1 Report

The comments are included in the attached Word file.

Reviewer 2 Report

The authors have worked on the removal of acrylic acid and acetic acid using two commercial Reverse Osmosis Membranes. The water flux, retention of the acids, the effect of concentration and influence of immersing the membranes in AA and HAc have been investigated in detail. There are some major concerns that have to be addressed. 

1) The specifications of the membranes used must be provided, along with the differences between the two membranes. 

2) The experimental section regarding the experiments conducted is completely missing. This is merged with the results section which makes it difficult to comprehend.

3) These are commercial RO membranes capable of removing NaCl >99.7- 99.8 %. So the reason for lower rejection of AA and HAc must be provided.

4) The "octanol/water partitioning coefficient" in Section 3.1.1 is incorrect to explain the observed phenomena. The references 25,26 give no insight about this. 

5) The AFM images and the XPS data can be included in the text as there are no limits to the number of fig. (If there are they must be made available in the supplementary information).

6) As GC has been employed to quantify HAc and AA. These also must be provided. At least for the initial study.

7) The pH of the experiments should be mentioned and these should be within the manufacturer specification.

8) Comparison with the latest state-of-the-art literature must be carried out.

9) The FTIR of the membranes immersed in the 7.5% acid solution could provide more insight into the degradation of the PA layer.

10) A large loss in the reduction of retention ability by the membranes could be an indication that the chosen membranes were not suitable for the particular application. Justification of the selection of the membranes must be included in the introduction section.

Round 2

Reviewer 1 Report

Please include the answer provided in relation to temperature in the manuscript:

"In the absorption process of acrylic acid production, the mixed gases produced in the oxidation process are introduced into the bottom of the absorption tower. Acrylic acid and multiple by-products are absorbed by the water sprayed down from the top of absorption tower. Then, the acrylic acid aqueous solution can be obtained at the bottom. The temperatures in the top and bottom of the absorption tower are about 43 °C and 70 °C, respectively. The temperature of the solution taken from the absorption tower is about 45 °C. According to the operation conditions of the membrane recommended by the manufacturer, the maximum operating temperature of the two RO membranes is 45 °C. Therefore, the solution taken from the absorption tower is required to cool down before entering into the RO unit."

Regarding the SD model, have the authors considered the osmotic pressure of the solution? I think it can not be neglected. Reformulate the fitting of the equation taking the osmotic pressure Δπ into account.

Author Response

Thanks, please see the attached response.

Reviewer 2 Report

The comments have been successfully incorporated into the revised manuscript.

Author Response

thanks